# Multifunctional Polymer Coatings of Fusible Interlinings for Sewing Products

**Nadezhda Kornilova** [1,*], **Albina Bikbulatova** [2,*], **Sergey Koksharov** [3], **Svetlana Aleeva** [3], **Olga Radchenko** [1] **and Elena Nikiforova** [1]

[1]  Engineering Center for Textile and Light Industry, Ivanovo State Polytechnic University, Sheremetevsky Ave. 21, 153000 Ivanovo, Russia; sva@isc-ras.ru (O.R.); nikiforova@ivgpu.com (E.N.)

[2]  Institute of Industrial Engineering, Information Technology and Mechatronics, Moscow State University of Food Production, Volokolamskoe Highway, 11, 125080 Moscow, Russia

[3]  Laboratory of Chemistry and Technology of Modified Fibrous Materials, G.A. Krestov Institute of Solution Chemistry of the Russian Academy of Sciences, Akademicheskaya st, 1, 153045 Ivanovo, Russia; nkorn@ivgpu.com (S.K.); bikbulatovaaa@mgupp.ru (S.A.)

*  Correspondence: nkorn@mail.ru (N.K.); albina-bikbulatova@yandex.ru (A.B.); Tel.: +7-905-107-69-89 (N.K.); +7-919-727-41-80 (A.B.)

**Abstract:** The aim of the study was to improve the range of adhesive interlining materials for clothes to add to the product a complex of improved consumer (shape stability, wear resistance) and special (protective, health-improving) properties. We have found that the modification of the adhesive coating of interlining materials with oligoacrylates provides a transition from the traditional discrete 2D glue interlayers between the bonded materials formation to the highly branched 3D structures of the interfacial layer of composites creation. Using dynamic light scattering method, IR spectroscopy, differential scanning calorimetry, gas adsorption, optical and scanning probe microscopy, methods of textile materials science, we have identified technological approaches that ensure the polymer coating penetration into the intrafiber nanopore spaces of the textile layers. We have identified ways to control the elastic-deformation properties of composites in the modifying the adhesive interlining material process in order to design its properties for the requirements of various garment models. We found a unique possibility of using graftable oligoacrylate dispersion to stabilize the nanodispersed state of functional fillers (silica, nanoferrites) and increase the uniformity of their distribution in the composite structure.

**Keywords:** nanostructured polymer coating; polyacrylate dispersion; nanodispersed fillers; graft copolymers; composite parts of a garment

## 1. Introduction

Fusible interlining materials (FIM) are used in production of a wide variety of garments: men's suits and shirts, women's jackets, skirts and trousers, special-purpose clothes for protection from various weather- and workplace-related effects, etc. The main purpose of applying these materials is to properly shape a garment and ensure shape stability in wear [1,2].

FIM are textile materials coated with a thermoplastic polymer (adhesive) in the form of a powder or paste in such a way that it remains on the material surface in the form of dots [3]. The mechanism of action of thermoplastic adhesives consists in polymer melt interaction with a fibrous base. This ensures strong bonding of fabric layers and makes the fused fabric composite rigid by forming a polymer matrix between the layers. The process of manufacturing a garment with good shape stability properties consists of three consecutive stages:

- making a fabric composite by gluing flat pieces of the shell fabric (main material) and FIM together (fusing);

- stitching the garment pieces with together;
- shaping the finished composite garment during wet-heat treatment (WHT).

Fusible interlining materials differ from each other in the fabric base structure, nature of the thermally fusible polymer, its amount and deposition method. The rigidity of a composite normally depends on the fiber composition of the fabric base, its surface density, mutual orientation of the warp threads of the shell and interfacing, type and amount of the thermally fusible polymer [3–5]. If the size of adhesive dots and their density increase, it results in effective growth in the fused fabric composite rigidity [2,5,6]. The higher rigidity makes it more difficult to shape the garment, and higher glue content leads to deterioration of hygienic properties, such as breathability.

The properties of FIM are selected considering the designed garment shape and the type of the main material (MM). The more rigid the garment form and the softer the MM, the higher bending deformation resistance fused fabric composite must have [7]. For example, the rigidity value (EI) of a fused fabric composite in different parts of a men's suit flap is required to range $(4.1–39.8) \times 10^{-3}$ N·cm$^2$. In small volume soft-plastic silhouette form garment models, the maximum EI values within one garment fabric piece can be 2.7 times higher than the minimum ones, whereas in the most rigid models of large volume, this difference equals 5.4 times [5]. In practice, shape stability is achieved by varying the number of interlining layers in different parts of a garment piece using several types of FIM with different rigidity and elasticity degrees. Among the disadvantages of this approach are the thick inner fabric layer that makes the garment heavier, the unpredictable deformation behavior of the garment in wear, poorer hygroscopicity and air permeability caused by the increase in the number of FIM layers and mass of the thermoplastic polymer. Additionally, it is quite natural that the very process of manufacturing of a composite garment piece increases its rigidity and lowers its shaping ability at the first production stage (fusing), which makes it more difficult to put the garment into the required shape at the third stage (final WHT).

An alternative to the use of FIM is the method of direct deposition of viscous polymer compositions in the form of a certain pattern onto the backside of the main material [3,8]. In such case, a polymer matrix is formed in the MM structure. Among the unquestionable advantages of the technique are the possibility of gradient reinforcement of a garment piece in order to regulate the properties of its separate parts, retention of the shaping ability by the composite fabric piece until the finishing WHT stage, and reduction in the garment weight. The main drawback is the risk of polymer penetration through the main material and spoiling of its outside surface appearance.

To ensure that the garment parts look the same, it is reasonable to introduce gradient changes in the stress-strain properties of the polymer matrix formed within the MM + FIM composite structure. This can be done through zonal deposition of a special polymer coating on the FIM surface. When selecting the coating for FIM surface modification, it is reasonable to employ the latest achievements in the field of synthesis of polymers with a complex spatial architecture. Quite promising are also methods of obtaining polymer matrix and interfacial layer in the form of molecular brushes and comb-like structures with numerous side chains attached to the backbone, which makes the macromolecule rigidity tens of times higher [9,10].

It is possible to additionally regulate the stress-strain properties of composite materials by using nanodispersed fillers with a high elasticity modulus, for example, silicon dioxide ($SiO_2$). One of the problems of polymer material modification by nanoparticles is related to their aggregation and nonuniform distribution within the polymer bulk [11,12]. Solving this problem will make it possible to provide fused fabric composites with additional properties by changing the filler type. For example, garments that have pieces with highly coercive nanoparticles in the fibrous base structure can have a positive effect on human health: improve the adaptation and regeneration capacity in stressful situations and protect the human body from high temperatures, acoustic, ultrasonic and electromagnetic exposure, impulsive loads and vibrations, as well as chemical and biological factors [13,14].

We have proposed a technology of FIM modification using a special polymer coating based on selecting a modifying polymer (MP) dispersion that can interact with a thermoplastic polymer (TP) forming a highly branched graft-copolymer, with the lateral branches penetrating into the pore system of the fibrous materials [15,16]. The conditions of producing copolymers of this type within the structure of the fused fabric composite formed must be suitable for MP and TP interaction and retention of the adhesive properties for TP and penetration into the fibrous component pores for MP. The formation of a highly branched polymer layer structure must not lead to an additional increase in the composite material rigidity before the garment takes its final shape [17]. The MP aqueous dispersion consistency must correspond to the conditions of screen printing on the FIM surface, and the MP dispersion degree must be high enough for the particles to diffuse into the inner volume of the fiber during drying process [18].

This paper presents the results of studying the possibility of regulating the stress-strain and consumer properties of composite garment parts by changing the conditions of polymer coating modification in order to obtain highly branched 3D interface structures. Special attention is paid to the degree of the modifying dispersion penetration into the fibrous material and introduction of nanodispersed fillers into the polymer matrix.

## 2. Materials and Methods

Five types of suite fabrics were used as the main materials in the study (Table 1).

**Table 1.** Characteristics of the main materials.

| Symbol | Fibrous Composition (%) | Surface Density $M_S{}^{MM}$ (g/m$^2$) | Stiffness EI$_{MM}$, $10^{-3}$ (N·cm$^2$) | | Shaping Ability $A_{MM}$ (%) | | Air Permeability $Q_{MM}$ (dm$^3$/s m$^2$) |
|---|---|---|---|---|---|---|---|
| | | | Warp | Weft | Warp | Weft | |
| MM1 | viscose 55, wool 35, polyester 10 | 240 ± 3 | 4.17 | 3.11 | 30.3 | 33.1 | 183 |
| MM2 | viscose 50, polyester 50 | 185 ± 2 | 2.3 | 2.36 | 35.3 | 34.2 | 275 |
| MM3 | viscose 80, polyester 20 | 190 ± 2 | 2.0 | 2.0 | 36.1 | 36.1 | 261 |
| MM4 | wool 53, polyester 44, elastane 3 | 167 ± 3 | 2.76 | 2.29 | 34.0 | 34.9 | 248 |
| MM5 | wool 99, elastane 1 | 167 ± 3 | 5.6 | 3.3 | 26.4 | 33.5 | 218 |

For modification, we used standard FIM, based on weft knit with polyamide adhesive dots on one side of it (Table 2).

**Table 2.** Characteristics of standard FIM.

| Symbol | Manufacturer | Fibrous Composition (%) | Surface Density $M_S$ (g/m$^2$) | Weft Threads Mass Fraction $G_{WT}$ (%) | Adhesive Coating Area [1], $S_{TP}$ (%) |
|---|---|---|---|---|---|
| FIM1 | Shanghai Uneed Textile Co.,Ltd, Shanghai, China | polyester 30, viscose70 | 80 | 60.8 | 13.2 |
| FIM2 | Shanghai Uneed Textile Co.,Ltd, Shanghai, China | polyester 30, viscose 70 | 65 | 54.4 | 16.7 |
| FIM3 | Kufner Textile Group, Unterhaching, Germany | polyester 27, viscose 73 | 58 | 70.5 | 18.7 |
| FIM4 | Iskozh JSC, Neftekamsk, Russia | polyester 60, cotton 40 | 75 | 60.8 | 25.2 |
| FIM5 | Iskozh JSC, Neftekamsk, Russia | polyester 100 | 70 | - | 15.8 |

[1] The adhesive coating area was determined by the number of adhesive dots in 1 cm$^2$ of FIM multiplied by the average area of one adhesive dot and divided by 100.

Samples of aqueous oligoacrylate dispersions Akremos (LLC "Pilot Plant of Acrylic Dispersions", Dzerzhinsk, Russia), Akratam AS and Anzal ("Pigment" Ltd., Tambov, Russia) with the nonvolatile substance content from 30 to 50 wt.% were used as the MP.

Two types of mechanical effects were applied to achieve particle disaggregation in the hydrosols: ultrasound treatment in a UZDN-2T disperser (LLC "U-RosPribor", Belgorod, Russia) at a frequency of 22 kHz and a combination of high shear stress, ultrasound and cavitation on a rotary-pulse activator (RPA) at the shear rate of $(0.5–17.4) \times 10^4$ s$^{-1}$.

Hydrosols of detonation nanodiamonds (DND) (Ioffe Physico-Technical Institute, St. Petersburg, Russia) and colloidal silicon dioxide (SD) acted as the nanodispersed filler (Guangzhou Jiechuang Trading Co. Ltd., Guangzhou, China; $SiO_2$ content 25%–26%, purity 99.5%). The DND was prepared by the oxidative synthesis method and had a zeta-potential of $-50$ mV and solid phase content of 3.62 wt.%.

The two-component MP systems with DND or SD were obtained by mixing 100 mL of an ultrasonically dispersed filler with 10 mL of a mechanically activated MP dispersion or by treating the mixture of the initial compounds with the required ratio of components in the RPA.

The size of the hydrosol nanoparticles was measured by the dynamic light scattering method on a Zetasizer Nano ZS analyzer (Malvern Panalytical, Malvern, UK); the signal accumulation time in a series of three measurements was 20 min. The analysis of the measurement results was carried out by an automated program based on the solution of the Fredholm integral equation of the first kind with an exponential kernel for the normalized correlation function [19]. To increase the recording ability of the measuring system, taking into account the recommendations [20] for the study of polyfraction systems in the results processing window, the value of the Lower Threshold indicator "0.05" must be corrected to "0" in accordance with the recommendations for studying polyfractional systems.

The MP interaction with the thermally fusible polymer was evaluated by the methods of IR-spectroscopy on an AVATAR 360 FT-IR ESP Fourier transform IR spectrometer (SpectraLab Scientific Inc., Markham, ON, Canada) and differential scanning calorimetry on a DSC 204 F1 Phoenix apparatus (NETZSCH, Selb, Germany) equipped with a μ-sensor. The two-component objects were prepared by introducing 10 wt.% of a powdered thermally fusible polymer into an MP hydrosol, then casting the samples on glass templates and drying them in air.

To develop the microporous structure of the FIM5 textile base polyester fiber, we carried out surface saponification in a boiling NaOH solution (0.25–1 mol/L) in the presence of Alcamon OS-3—a quaternary ammonium compound (0.3 g/L). The processing duration was varied from 3 to 10 min and was followed by washing with cold and hot (80 °C) flow-through water until the phenolphthalein reaction became neutral. The effectiveness of the fiber surface modification was evaluated by the low-temperature nitrogen adsorption-desorption method at 77 K on a NOVA 1200e gas sorption analyzer (Quantachrome, Boynton Beach, FL, USA) to find out the material porosity (VP, $m^3/g$) and inner pore size distribution.

The area of the FIM surface covered with the modifying dispersion—SA—was varied within the range from 0.35 to 0.65 by changing the screen patterns that had different density and line thickness values. The patterns of MP dispersion deposition and the respective values of the relative interlining reinforcement area are shown in Table 3.

**Table 3.** Characteristics of patterns for applying MP.

| Number | 1 | 2 | 3 | 4 | 5 |
|---|---|---|---|---|---|
| Pattern of MP dispersion deposition | 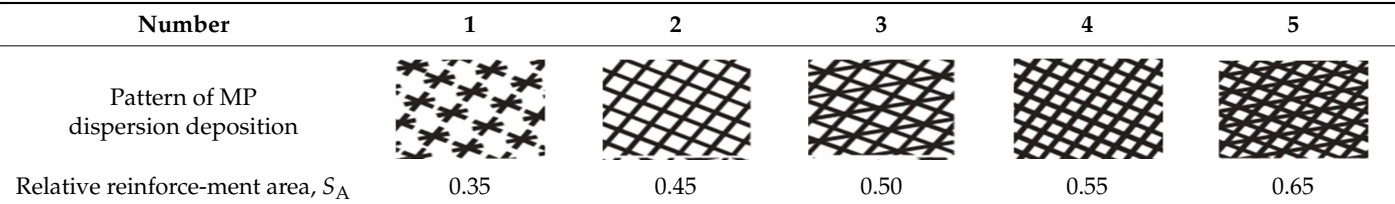 | | | | |
| Relative reinforce-ment area, $S_A$ | 0.35 | 0.45 | 0.50 | 0.55 | 0.65 |

The fabric composites were made by fusing the main material with a standard or modified FIM on a Japsew SR-600 press (Japsew Corporation, Shanghai, China) a temperature of 110 °C for 20 s. The wet-heat treatment of the fused fabric composites was carried out on a Malkan UPP1AVK (Malkan Machinery, İstanbul, Turkey) press at a temperature of 140 °C for 30 s. The wetting reached 20%–30%.

The processing characteristics and consumer properties of the fused fabrics and finished composites with standard and modified FIM were estimated by the following

indicators: thickness, stiffness, elasticity, shaping ability, bond strength, hygroscopicity, air permeability, and shape stability factor.

The indicator thickness was conducted in accordance with the ISO 5084:1996 standard.

The indicator stiffness (EI) of textile liner and fused panel was calculated in accordance with GOST RF 10550-93 using contactless console methodology. The strips of material size of $160 \times 30$ mm$^2$ were used. They were placed horizontally on the top side of the machine and pressed by the load size of 20 mm in the middle. After that, the sides of anchor were dropped down, the ends of material hanged loose due to gravitational force. After a minute, the amount of overhang of the strips ends (*f*) was measured. The value of stiffness EI ($10^{-3}$ N·cm$^2$) was calculated using following equation:

$$EI = \frac{42.046 \times m}{0.5755 \times f^3 - 2.411 \times f^2 + 8.502 \times f}$$

where *m* is the mass of material strip in grams.

The indicator elasticity (*U*) was calculated using the ring method in accordance with GOST RF 10550-93 using ring methodology. A sample $95 \times 20$ mm$^2$ was clamped on the removable site. The top surface of the sample was outside. The sample should take the form of correct ring. Then it was loaded up to the sample sagging was 1/3 of diameter ($S_0 = 30$ mm) during 30 s. After load breaking the sample laid off during 30 s and the residual deflection $S_1$ was measured. The value of indicator *U* (%) was calculated using following equation:

$$U = \frac{S_0 - S_1}{S_0 \times 100\%}$$

The indicator shaping ability (*A*) characterizes the relative size of the material (composite) field which repeats the three-dimensional surface. It was conducted in accordance with patent RF No.234347 "The method of measurement molding ability of textile liner". For determination of molding ability, the sphere with diameter 150 mm was set on the tripod, which was fixed on the stand. The sample of material the size $350 \times 350$ mm$^2$ was placed from above the sphere. For ensuring a fit of the material to the sphere, folds of the same depth were made in the warp and weft directions. The indicator "shaping ability" (*A*) was calculated using following equation:

$$A = \frac{\alpha}{\alpha_{max}}$$

where $\alpha$ is a central angle of recurrence by material (composite) the surface of sphere, degrees; $\alpha_{max}$ is a maximum possible angle of recurrence ($\alpha_{max} = 1800$).

Methods for determining indicators EI, *U*, *A* are detailed in the article [17].

The indicator bonding strength (*P*) was conducted in accordance with GOST USSR 28832-90; sample preparation and conducting tests were realized in accordance with the ASTM D 2724 standard under the following conditions: sample size was $150 \times 30$ mm$^2$, size of unglued piece was 40 mm, the distance between clamps was 50 mm, the speed of moving motile clamp was 100 mm/min. The indicator bonding strength (*P*, N/10 cm) was calculated using equation:

$$P = \frac{F}{300}$$

where *F* is average value of the force required to separate the interlining and shell fabric.

Indicator hygroscopicity was conducted in accordance with ISO 811-81 Textile fabrics, Methods for determination of hygroscopic and water-repellent properties.

Indicator air permeability (*Q*) was conducted in accordance with GOST 12088-77. The methodology corresponds to ASTM D737 Standard Test Method for Air Permeability of Textile Fabrics. Measurements were calculated under the following conditions: rarefaction

under dotty sample 49 Pa and jaw level for dotty sample 147 N. Air permeability $Q$ (dm$^3$/s·m$^2$) was calculated using the equation:

$$Q = \frac{V}{S}$$

where $V$ is average volume of air, dm$^3$/s; $S$ is measurement area, m$^2$.

The resistance of the composite shape to wear (storage, loading, axial deformation, dry cleaning) was measured on hemispherical samples with a 10 cm radius formed on a laboratory press from fused fabrics in recommended WHT modes. The samples were stored in a free state at room temperature. They were exposed to static loading for 10 min with a 30 g load (Figure 1), followed by a 5 min rest.

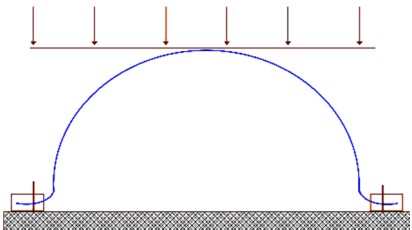

**Figure 1.** Loading scheme for bulk samples.

The deformation of the hemispherical samples was achieved by stretching them in two perpendicular radial directions at the same rate using a specially developed device that made it possible to mechanize the deformation in this mode creating the necessary conditions for objective studies of the tensile strain. The number of deformation cycles was 20,000. After the deformation, the samples were released from the apparatus clamps and were left in a state of rest for 10 min.

The shape stability factor (SS, %) was calculated by the equation:

$$\text{SS} = \frac{H_i}{H_0} \times 100\%$$

where $H_0$ and $H_i$ are the sample height values 10 min after the sample was formed and after the exposure.

### 3. Results

*3.1. Justification of the Conditions of Two-Component Polymer Coating Formation*

Modified polymer coatings can be obtained for a lot of different FIM with a wide variety of thermoplastic compounds (polyolefins, aliphatic and aromatic polyamides, polyethers and polyesters, polyvinyl chloride, polyurethanes, polyvinyl acetate, ethylene, and vinyl acetate copolymers or acrylic compounds) as the thermoplastic adhesive polymer coating (TP). The modifying compound was selected based on its ability to interact with the TP within the range of temperatures that are used in the fusing processes of sewing production, and to ensure that the TP did its job properly. The best results for the most common in garment production polyamide TP were obtained using polyacrylates as the MM. Figure 2 presents the results of controlled interaction of the polyamide adhesive PA-12AKR and oligoacrylate dispersion Akremos 120D as an example.

Using the IR spectroscopy study results (Figure 2a), it was possible to control the type of interactions taking place during the formation of the polyamide–acrylate copolymer under heat treatment. The narrowing of the absorption band (at 725 cm$^{-1}$ in the adduct spectrogram) of the N–H bond stretching vibrations in the polyamide compound amino group was accompanied by the disappearance of the peak at 942 cm$^{-1}$, corresponding to the double bond stretching vibrations in the CH2=C oligoacrylate molecule. At the same time, the high-intensity absorption on the stretching vibration bands in the C=O and

N–H polyamide chain groups remained the same—860 and 1730 cm$^{-1}$, respectively, which ensured fabric layer fusion.

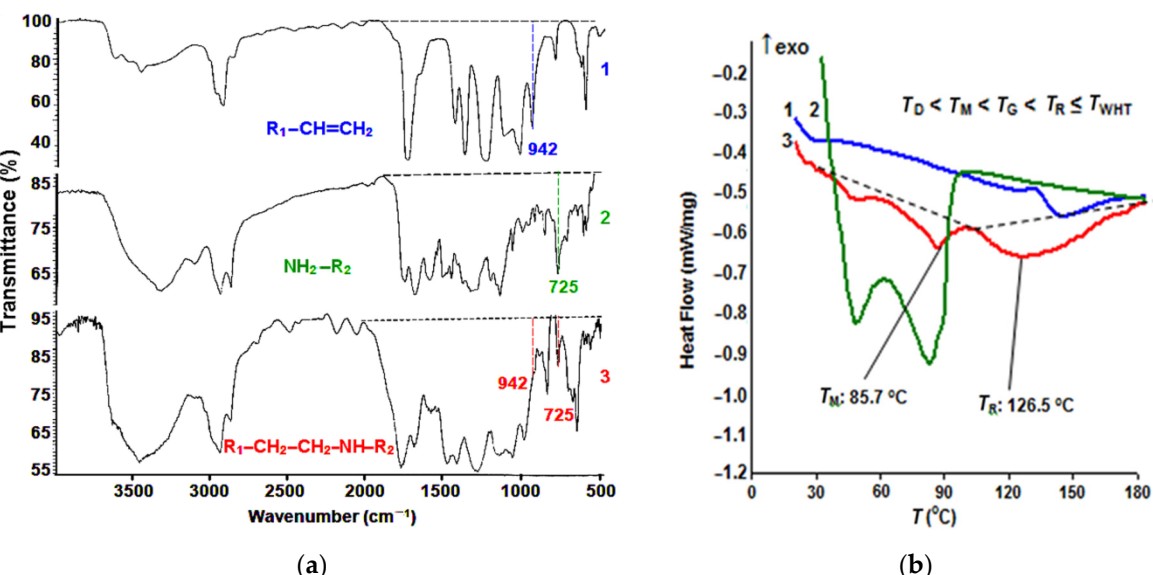

**Figure 2.** IR spectra (**a**) and DSC curves (**b**) of Acremos 120D oligoacrylate dispersion (1), PA-12AKR polyamide adhesive (2), and polyamide-polyacrylate adduct (3).

The DSC data shown in Figure 2b made it possible to determine the correspondence of the copolymerization temperature peak $T_R$ = 126.5 °C with the WHT heating parameters. To create the optimal manufacturing conditions, the preliminary fusion of the MM and FIM on a thermal press must be carried out at room temperature exceeding the TP melting point in the MP presence ($T_M$ = 85.7 °C), but by no more than 25 °C [17] to prevent considerable spreading of the adhesive dots and, thus, the rigidity increase in the fused fabric.

By analyzing the data in Figure 2, we determined the most suitable modes of the main composite preparation stages. It is proposed to deposit an MP dispersion onto the FIM side coated with an adhesive layer by the screen-printing method, dry it at a temperature below the TP melting point, and then obtain a modified fusible interfacing material (MIM). The following parameters are required to create a fusible coating based on the polyamide PA-12 AKR: MIM drying temperature $T_D$ = 60 °C, MM and MIM fusing temperature $T_G$ = 110 °C, and temperature of the finished garment WHT $T_{WHT}$ = 140 °C.

Table 4 presents measurement results of the bond strength of FIM + MM or MIM + MM fused fabrics obtained at $T_G$ = 110 °C and subjected to WHT at $T_{WHT}$ = 140 °C. The data in Table 4 indicate that the copolymer retained its adhesion properties. The bond strength of MM + MIM fused fabrics was 15%–65% higher than the respective value in the initial standard FIM. The cohesive bond breakage became adhesive, with fibers pulled out of the textile base structure, which indicates that the fibrous materials actively participated in the formation of the highly developed composite interface.

In contrast to the traditional method of formation of a 2D-structured adhesive interlayer that penetrates to a small depth in the interfiber spaces on the surface of the composite layers, graft-copolymers must be introduced into the capillary and pore system of individual fibers. In order to achieve that, a graft polymer dispersion must penetrate into the interfiber pore spaces of the interlining fabric base. It should be borne in mind that swelling of cellulose fibers increases the lateral size of mesoporous cavities to 25–35 nm, with the submicroscopic pore diameter reaching 3–7 nm.

The ability of industrially produced polymer dispersions to penetrate into the structure of the FIM textile base was evaluated using the results of a study of hydrosol particle sizes with the dynamic light scattering method. One of the few compounds satisfying the size

requirements was Akratam AS 01, an acrylate dispersion preparation. The characteristics of its colloidal state are shown in Figure 3.

**Table 4.** The bond strength of fused fabrics.

| Fused Fabrics | | Bond Strength, *P* (N/10 cm) | | $\Delta P$ (%) |
|---|---|---|---|---|
| MM Type | FIM | MM + FIM | MM + MIM | |
| MM1 | FIM3 | 4.7 | 7.1 | 51 |
| | FIM2 | 4.1 | 5.8 | 41.5 |
| MM2 | FIM1 | 3.7 | 5.4 | 46 |
| | FIM4 | 5.5 | 7.7 | 40 |
| MM3 | FIM3 | 4.5 | 6.6 | 46.7 |
| | FIM4 | 5.3 | 7.8 | 47.2 |
| MM4 | FIM2 | 4.0 | 5.6 | 40 |
| | FIM1 | 3.6 | 5.1 | 41.7 |
| MM5 | FIM2 | 4.3 | 6.4 | 48.8 |
| | FIM4 | 5.4 | 7.7 | 42.6 |

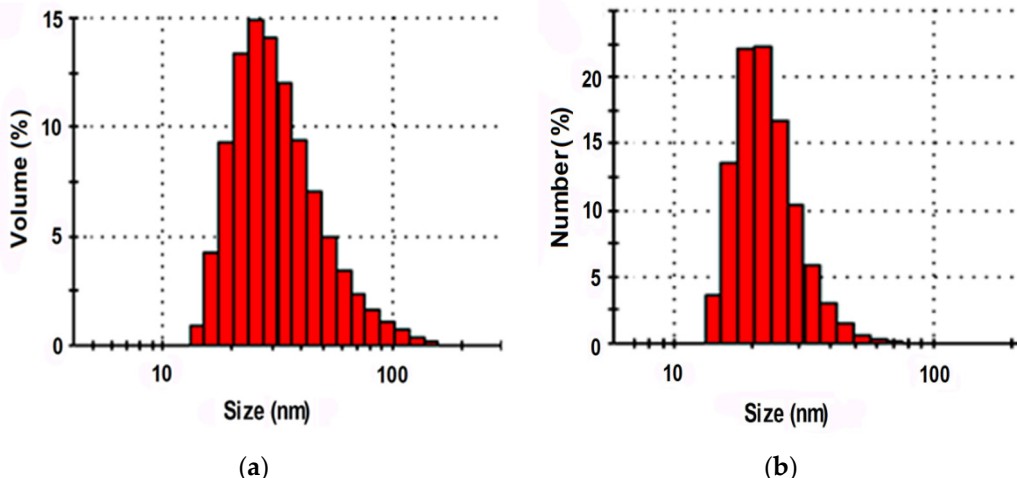

(**a**)  (**b**)

**Figure 3.** Particle size distribution of the relative dispersed phase volume indicators (**a**) and the particles' relative number (**b**) in the preparation Akratam AS 01 hydrosol.

It is shown that more than 86% of the relative particle number and about 70% of the relative dispersed phase volume were made up by fractions with particles less than 35 nm in size. It means that the compound could migrate into the structure of the cellulose base when the fibrous material gets swollen.

Table 5 shows the results of the particle size analysis of all studied acrylate compounds.

In the vast majority of the studied objects, the size of the particles ($r^{MF}$) of the major fractions, identified by the total value of the dispersed phase relative volume ($V^{MF}$) or relative number of particles ($N^{MF}$), was much bigger than those of the voids in the cellulose fiber capillary pore system. When such compounds are used in their initial form, the dispersed phase particles form agglomerates in the interfiber and interthread spaces of the FIM textile base, which lowers the effectiveness of an adhesive polymer coating modification. The advantages of using compounds with a large share of small-sized fractions consist in the possibility of formation of elongated graft chains in the fibrous material capillaries and the increase in the density of grafting of the side branches to the polymer backbone.

It has been established that it is possible to effectively increase the dispersion degree of oligoacrylate compounds by subjecting them to a series of mechanoacoustic effects: high shear stresses, ultrasound, and cavitation. For example, Figure 4 illustrates changes in the Akremos 120D particle sizes after their processing on a rotary-pulse activator (RPA).

**Table 5.** Estimation results of the major fractions particlesize ($r^{MF}$) and relative share ($V^{MF}$) in hydrosols of MP dispersions.

| Type of MP Dispersion | V = f (r) Curve Data | | N = f (r) Curve Data | |
|---|---|---|---|---|
| | $r^{MF}$ (nm) | $V^{MF}$ (%) | $r^{MF}$ (nm) | $N^{MF}$ (%) |
| Akratam AS 01 | 21.9–45.6 | 78.4 | 18.9–34.0 | 86.7 |
| Akratam AS 01-M | 25.4–45.6 | 41.5 | 21.9–34.0 | 78.3 |
| | 171–413 | 26.3 | - | - |
| Akratam AS 02 | 99.1–171 | 82.5 | 70.9–128 | 85.4 |
| Acremos 120Д | 32.9–110 | 82.5 | 31.6–82.1 | 87.4 |
| Acremos 304 | 82.1–148 | 84.0 | 70.9–128 | 86.9 |
| Acremos 306 | 82.1–48 | 81.8 | 70.9–128 | 83.0 |
| Acremos 402 | 52.9–110 | 73.5 | 45.6–82.1 | 85.6 |
| Anzal KS | 198–266 | 96.1 | 198–266 | 94.4 |

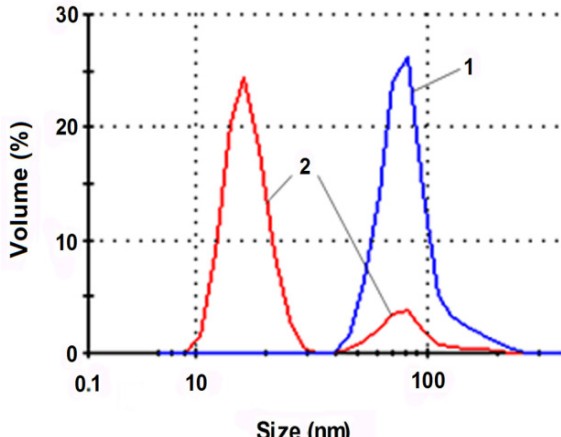

**Figure 4.** Particle size distribution of the dispersed phase relative volume in the initial dispersion of Acremos 120D (1) and after mechanoacoustic treatment on the RPA apparatus (2).

Hydrosol mechanical activation shifts the peak related to the particle hydrodynamic radius, rmax, from 75 to 15 nm. More than 80% of the dispersed phase relative volume is taken up by the fractions with particles less than 35 nm in size that can be absorbed by the mesoporous cavities of the cellulose fiber. Mechanical activation of the Akratam AS 01 aqueous dispersion produces an ultradispersed hydrosol form with $r_{max}$ = 2.5 nm.

The MP particle size effect on the properties of the fused fabric MM3 + MIM2 composite is illustrated in Figure 5. The rigidity of the composite was compared with that of the fused MM3 + FIM2 composite. Deposition of an oligoacrylate dispersion in its initial form (curve 1) led to a two-fold increase in the $EI_K$ value at the MP content on the material of 0.3 wt.%. Ultrasound treatment aimed at dispersion disaggregation reduced the size of the MP particles to 40 nm (curve 2), which, however, did not make them small enough to penetrate inside the fiber. In this case, 3D-copolymer structures were also formed in the interthread and interfiber spaces of the textile base. Additionally, the $EI_K$ gain achieved by grafting smaller radicals was 17% lower than that reached by deposition of the same amount of the MP in its initial form (see Figure 5).

When MP particles were subjected to mechanically activated grinding and were reduced in size to the mesoporous spaces of the swollen cellulose fiber (curve 3), the rigidity gain ($\Delta EI_K$) relative to that in the initial FIM ($G_{MP}$ = 0) was 3–3.5-fold in comparison to that achieved by deposition of the same amount of the nonactivated dispersion. When ultradispersed MP particles were used (curve 4), they occupied the whole inner volume of the fiber, including the submicroscopic pores, which led to an up to 10-fold increase in $\Delta EI_K$.

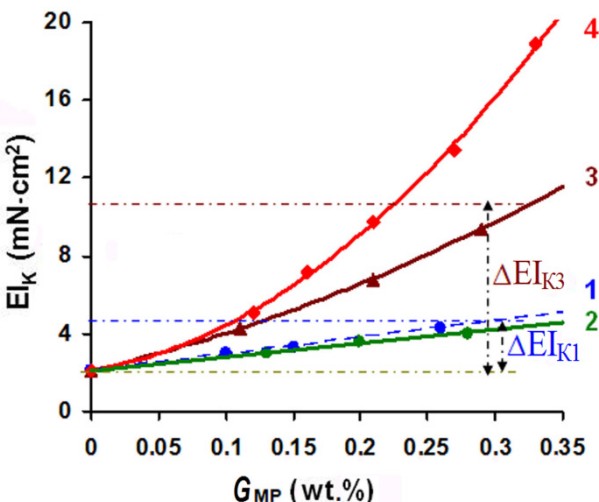

**Figure 5.** MM3 + MIM2 composite stiffness index ($EI_K$) dependence on the applied modifying polymer dispersion amount ($G_{MP}$) with the value of the hydrodynamic particle size $r_{max}$: 1–75 nm; 2–40 nm; 3–15 nm; 4–2.5 nm.

The data in Figure 5 allow us to describe the concentration dependencies of the rigidity gain with the following expressions (the value in brackets shows the MP dispersion particle size):

$$\Delta EI_{(75nm)} = 8.38 \times G_{MP};$$
$$\Delta EI_{(15nm)} = 18.88 \times G_{MP} + 7.36 \times G^2_{MP};$$
$$\Delta EI_{(2.5\ nm)} = 11.39 \times G_{MP} + 117.83 \times G^2_{MP} \quad (1)$$

The dependencies presented reflect the accelerating increase in the rigidity due to the effect of the FIM textile base pore system. Thus, it was reasonable to apply a combination of mechanoacoustic effects to reduce the size of the oligoacrylate particles to 2.5–30 nm to ensure their penetration into the FIM pores.

In order to achieve similar effects on synthetic fabrics (polyester and polyether in particular), which are chemically inert, have a small number of active groups on the surface, are smooth, and have no intrafiber voids, we suggested conducting surface saponification of polyethylene terephthalate in the presence of an interphase catalyst ensuring hydrolysis localization and formation of numerous nano- and microcavities. The role of the catalyst can be played by some quaternary ammonium compounds, in particular Alkamon OS-3.

Figure 6 shows the effects of the saponification conditions on changes in the polyester fiber-free volume ($V_P$) and stiffness ($EI_K$) of the composite after fusing and WHT.

As Figure 6 shows, changing the conditions of the polyester fiber saponification stage in the textile base and the amount of the Akratam AS 01 dispersion deposited on the substrate was an effective method of composite rigidity increase control. In comparable experimental conditions, when the compound was used in its initial form and could fill mesoporous cavities, the stiffness gain ($\Delta EI_K$) caused by depositing equal MP amounts could range from 2.3 to 10 times. In the case of the ultradispersed MP form that can penetrate the cavities of submicroscopic sizes, the $\Delta EI_K$ value could increase 12.7–16.4 times. The best type of fiber preactivation for treating the initial form of the Akratam AS 01 polymer dispersion was mode 3 ($C_{NaOH}$ = 1 mol/L, $\tau$ = 3 min). In case of the ultradispersed MP form, surface hydrolysis could be carried out at an alkali concentration lowered to 0.25 mol/L.

### 3.2. Evaluation of the Polymer Coating Effect on the Processing and Consumer Characteristics of the Obtained Composite

An evaluation of the processing and consumer properties of the composite materials for garment shaping pieces after WHT (finished composite) and semifinished composites

at the intermediate stage of fusing (fused fabric) was done. Some of the results for the samples obtained using standard FIM*i* and their modified forms (MIM*i*) are shown in Tables 6 and 7. The MIM samples were prepared on a relatively small area that was coated with the ultradispersed preparation Akremos 120D ($S_A$ = 0.45). The weft threads of the main material and the interlining were aligned with each other during the fusing.

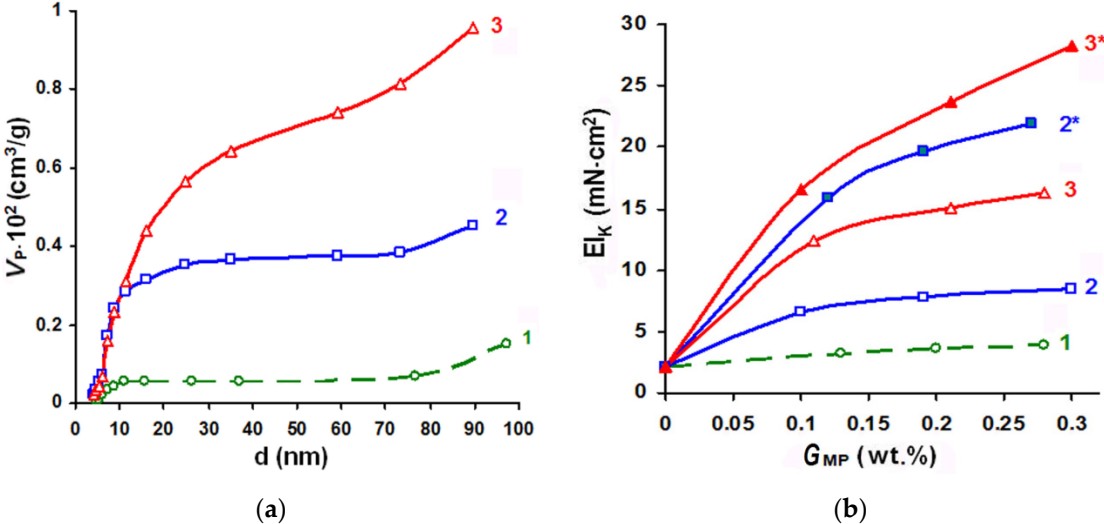

(**a**)                                                           (**b**)

**Figure 6.** Influence of TPM5 saponification conditions on the dependence of the free volume distribution over the pore diameter (**a**) and changes in the composites stiffness (**b**) with varying the amount of MP applied in the initial (1–3) and ultradispersed (2*–3*) forms: 1-standard TPM5; saponification conditions: 2, 2*-$C_{NaOH}$ = 0.25 mol/L, $\tau$ = 10 min; 3, 3*-$C_{NaOH}$ = 1 mol/L, $\tau$ = 3 min.

**Table 6.** Processing properties comparative characteristics (stiffness EI, forming ability *A* and elasticity *U*) of fused fabric (index "FF") and finished composite (index "K") obtained using FIM and MIM.

| Composite | | $EI_K$ (10$^{-3}$ N·cm$^2$) | | $EI_{FF}$ (10$^{-3}$ N·cm$^2$) | | $A_{FF}$ (%) | | $U_{FF}$ (%) | | $U_K$ (%) | |
|---|---|---|---|---|---|---|---|---|---|---|---|
| | | Warp | Weft | Warp | Weft | Warp | Weft | Warp | Weft | Warp | Weft |
| MM1+ | FIM1 | 12.1 | 13.4 | 8.2 | 9.4 | 27.1 | 24.7 | 39.5 | 46.4 | 58.6 | 68.0 |
| | MIM1 | 23.9 | 25.5 | 8.3 | 9.7 | 27.2 | 24.4 | 39.7 | 46.2 | 79.5 | 88.2 |
| MM1+ | FIM3 | 9.4 | 17.1 | 6.6 | 11.5 | 28.5 | 27.4 | 39.4 | 48.5 | 56.2 | 72.1 |
| | MIM3 | 21.2 | 28.5 | 6.3 | 11.1 | 28.7 | 27.5 | 39.1 | 48.6 | 77.2 | 91.6 |
| MM1+ | FIM4 | 14.8 | 23.4 | 9.7 | 14.3 | 26.6 | 21.3 | 42.0 | 51.2 | 61.8 | 74.7 |
| | MIM4 | 26.1 | 35.5 | 9.5 | 14.5 | 26.3 | 21.5 | 41.8 | 51.0 | 80.3 | 94.2 |
| MM4+ | FIM1 | 9.1 | 9.4 | 6.1 | 6.5 | 32.6 | 29.5 | 35.5 | 46.0 | 50.0 | 57.1 |
| | MIM1 | 21.2 | 21.4 | 5.8 | 6.3 | 32.8 | 29.5 | 35.1 | 45.7 | 72.5 | 78.6 |
| MM4+ | FIM3 | 7.6 | 12.6 | 5.1 | 8.5 | 32.9 | 31.0 | 34.3 | 39.1 | 49.0 | 55.8 |
| | MIM3 | 19.4 | 23.8 | 4.9 | 8.6 | 32.7 | 31.2 | 34.2 | 39.1 | 71.5 | 77.3 |
| MM4+ | FIM4 | 10.8 | 16.7 | 7.1 | 11.1 | 30.1 | 27.8 | 37.3 | 42.8 | 53.9 | 62.0 |
| | MIM4 | 22.2 | 27.8 | 7.3 | 10.7 | 30.3 | 27.6 | 37.2 | 43.1 | 75.4 | 82.5 |

An analysis of the initial parameters of the objects to be modified considering the processing characteristics of the FIM, given in Table 2, was done. In all of the suite fabrics, we observed the same types of regularities, namely an increase in the stiffness and elasticity of the composite along the weft aligned with the reinforcing weft yarn in the FIM structure. The $EI_K$ and $U_K$ values also became higher when the mass of the textile base and the $M_S$ and $S_{TP}$ indicators of thermoplastic adhesive were increased. The 1.9-fold growth in the adhesive coating area following the replacement of FIM1 with FIM4 led to a 1.75-fold stiffness increase along the weft despite a 7% decrease in $M_S$ value. Along the warp (when FIM did not have a reinforcing thread), the increase in stiffness along the warp value

following the $S_{TP}$ growth was smaller (1.22-fold) and more sensitive to changes in the interlining fabric mass.

**Table 7.** Consumer properties of composites based on FIM and MIM: thickness h; shape stability factor for storage in 24 h ($SS_{24}$), 48 h ($SS_{48}$), loading ($SS_{load}$), multiaxis cyclic tensile strain ($SS_{tensile}$), and dry cleaning ($SS_{dry\ clean}$); air permeability $Q$ and hygroscopicity Hyg.

| Composite | | h (mm) | Shape Stability Factor (%) | | | | | $Q$ (dm³/s m²) | Hyg (%) |
|---|---|---|---|---|---|---|---|---|---|
| | | | $SS_{24}$ | $SS_{48}$ | $SS_{load}$ | $SS_{tensile}$ | $SS_{dry\ clean}$ | | |
| MM1+ | FIM1 | 1.90 | 85 | 84 | 75 | 70 | 74 | 84.3 | 5.3 |
| | MIM1 | 1.86 | 90 | 89 | 81 | 77 | 80 | 83.9 | 5.4 |
| MM1+ | FIM3 | 1.76 | 87 | 85 | 76 | 72 | 76 | 78.1 | 5.5 |
| | MIM3 | 1.75 | 93 | 91 | 82 | 79 | 81 | 77.8 | 5.7 |
| MM1+ | FIM4 | 1.76 | 91 | 88 | 82 | 80 | 85 | 50.1 | 5.1 |
| | MIM4 | 1.75 | 96 | 94 | 88 | 83 | 91 | 50.1 | 5.1 |
| MM4+ | FIM1 | 2.0 | 77 | 73 | 64 | 61 | 63 | 108.5 | 4.4 |
| | MIM1 | 1.95 | 81 | 78 | 74 | 69 | 68 | 107.9 | 4.6 |
| MM4+ | FIM3 | 1.90 | 76 | 73 | 59 | 60 | 63 | 84.3 | 4.5 |
| | MIM3 | 1.85 | 80 | 78 | 73 | 67 | 69 | 84.4 | 4.7 |
| MM4+ | FIM4 | 1.80 | 80 | 78 | 69 | 65 | 70 | 52.1 | 4.2 |
| | MIM4 | 1.80 | 85 | 84 | 77 | 72 | 75 | 52.5 | 4.3 |

When standard FIM were used, the maximum $EI_K$ value was $23.4 \times 10^{-3}$ N·cm². To reach this value, the MM1 garment pieces had to be oriented crosswise, along the reinforcing weft thread of FIM4, the processing characteristics of which were close to the extremely high values: $M_S = 75$ g/m²; $G_{WT} = 60.8\%$; $S_{TP} = 25.2\%$. A further increase in the rigidity in order to fix the garment shape could be only achieved by using several FIM layers.

The main purpose of FIM polymer coating modification is increasing the $EI_K$ level in order to reduce the number of layers in the fused composite and lower materials' consumption. A comparison of the data in Tables 1 and 6 showed that the $EI_K$ gain relative to the main material rigidity ($EI_{MM}$) increased from 2.2–7.6 times in the fabrics with standard FIM to 5.1–11.5 times in the used MIM variants.

After the fusion stage, the rigidity of the standard fused fabric composites ($EI_{FF}$) became 1.5–4.8 times higher than that of $EI_{MM}$. The higher rigidity of semifinished product had a negative effect on their ability to take the required shape. In the samples with FIM, the decrease in the fused fabric shaping ability $A_{FF}$ in comparison with the $A_{MM}$ reached approximately 35.6%. In spite of the good structural mobility of knitted FIM, the use of molten adhesive dots to fix their parts on the MM surface made the fused composite elasticity quite high. The $U_{FF}$ value along the reinforcing weft threads of the FIM exceeded 50%. Since excessive elasticity of fabrics can be an obstacle to garment shaping from flat pieces, it is desirable to prevent its growth in a fused fabric.

This means that an important component of interlining polymer coating functionalization is ensuring the timeliness of copolymerization in order to preserve the maximum plasticity of the fused composite and obtain the necessary 3D shapes with their subsequent fixation at the final WHT stage. The data in Table 6 provides clear evidence that the use of finely dispersed graft forms and their introduction into the pore structure of the MIM textile base fibers have no effect on the stress-strain properties of the fused fabric. The $EI_K/EI_{FF}$ ratio value rose from 1.4–1.6 in FIM-based fused fabrics to 2.5–4 in the MIM-based ones.

An equally important aim of the interlining polymer coating functionalization is higher elasticity of finished composite ($U_K$) for shape stability under external effects. A comparison of the values of the $EI_K^{weft}$ and $U_K^{weft}$ indicators for FIM1 and FIM3 fabrics showed that the higher rigidity of the standard interlining materials did not always make their elasticity higher as a result of the action of several structural factors. The proposed technology of forming a highly branched composite interface with the side chains of the graft-copolymer penetrating into the intrafiber pore spaces of the fusing textile layers allowed a simultaneous increase in $EI_K$ and $U_K$ indicators. The relative elasticity gain after

the WHT ($U_K/U_{FF}$) rose from 1.2–1.5 times in the fused fabrics with FIM to 1.7–2.1 times in the samples with MIM.

Dependencies have been obtained showing how the processing parameters of the main material (the MM index) and FIM structural parameters—surface density ($M_S$, g/m$^2$), weft thread mass fraction ($G_{WT}$, %), and adhesive coating area ($S_{TP}$, %)—affect the changes in the respective properties of the composite obtained using standard FIM:

$$
\left\{
\begin{array}{c}
EI_K^{warp} = EI_{MM}^{warp} \times (0.029 \times M_S + 0.049 \times S_{TP}); \ R^2 = 0.979; \\
EI_K^{weft} = EI_{MM}^{weft} \times (0.005 \times M_S + 0.002 \times G_{WT} + 0.27 \times S_{TP}); \ R^2 = 0.982; \\
EI_{FF} = (0.67 \ldots 0.72) \times EI_K; \\
A_{FF}^{warp} = \frac{A_{MM}^{warp}}{(0.012 \times M_S + 0.017 \times S_{TP})}; \ R^2 = 0.987; \\
A_{FF}^{weft} = \frac{A_{MM}^{weft}}{(0.011 \times M_S + 0.01 \times G_{WT} + 0.323 \times S_{TP})}; \ R^2 = 0.984; \\
U_K^{warp} = 44.5 + 0.14 \times M_S + 0.26 \times S_{TP}; \ R^2 = 0.88; \text{—only for MM1} \\
U_K^{weft} = 45 + 0.0186 \times M_S + 0.095 \times G_{WT} + 0.65 \times S_{TP}; R^2 = 1; \text{—only for MM1} \\
U_{FF} = (0.65 \ldots 0.75) \times U_K
\end{array}
\right.
\tag{2}
$$

The data in Table 6 indicate that deposition of a modifying dispersion had practically no effect on the processing characteristics of the obtained fused fabric. Hence, Equation (2) used for $EI_{FF}$, $A_{FF}$, and $U_{FF}$ indicators can be also applied to MIM.

Table 7 shows the effect of the interfacing polymer coating modification on changes in the consumer properties of the obtained composite samples.

The composites with MIM were found to be thinner than those with FIM. It was, evidently, the result of the denser fiber structure of the interfacing material caused by MP penetration into the pores and formation of a developed interphase layer. This also made the shape stability factor (SS) of the obtained MIM-based finished composite noticeably higher both in storage and in wear. It should be said that the SS reduction on the second day of storage in both types of interlining materials was rather small, i.e., the shape relaxation took place in the first 24 h after the fabric was formed, and then the shape remained stable. The shape stability values for the composites correlated with the changes in their elasticity. In the FIM-based samples, the correlation between the indicators can be described by the following set of equations:

$$
\begin{array}{c}
SS_{24} = 0.71 \times U_K^{warp} + 0.64 \times U_K^{weft}; \ R^2 = 0.989; \\
SS_{load} = 0.7 \times U_K^{warp} + 0.5 \times U_K^{weft}; \ R^2 = 0.989; \\
SS_{tensile} = 0.5 \times U_K^{warp} + 0.62 \times U_K^{weft}; \ R^2 = 0.985
\end{array}
\tag{3}
$$

The shape stability factors for the MIM-based composites can be described by the following set of equations:

$$
\begin{array}{c}
SS_{24} = 0.071 \times U_K^{warp} + 0.96 \times U_K^{weft}; \ R^2 = 0.992; \\
SS_{load} = 0.36 \times U_K^{warp} + 0.61 \times U_K^{weft}; \ R^2 = 0.981; \\
SS_{tensile} = 0.14 \times U_K^{warp} + 0.76 \times U_K^{weft}; \ R^2 = 0.988
\end{array}
\tag{4}
$$

Hygiene safety of composite materials for making clothes is evaluated by breathability and hygroscopicity. Fabric breathability is important for maintaining the optimal temperature in the undergarment space and for diversion of carbon dioxide vapor intensively released by human skin (about 250 mg/h). The $Q$ indicator values for composite samples were 55%–58% lower than the initial level for the MM, as fabric bonding dramatically increases the resistance to the air movement. The increase in FIM surface density and TP mass was accompanied by a considerable reduction in the composite $Q$ value. The macropores in the composite structure were blocked by the adhesive interlayer more quickly as the

adhesive coating area $S_{\text{TP}}$ became larger. The correlation between the values is described by the equation:

$$Q = \frac{Q_{\text{MM}}}{0.019 \times M_{\text{S}} - 0.014 \times S_{\text{TP}} + 0.0042 \times S_{\text{TP}}^2}; \ R^2 = 0.987 \tag{5}$$

An important fact is that MP dispersion introduction into the fibrous material structure and formation of a graft-copolymer adhesive form in fused composites with MIM did not lead to additional resistance of the composite garment pieces to air penetration. At the same time, the effectiveness of the polymer coating modification could be evaluated by determining the $\text{EI}_{\text{K}}/Q$ indicator ratio, which becames 1.5–2.6 times higher when standard FIM were replaced with developed MIM.

Hyg indicator values are interrelated with the content and hydrophilicity of fibrous components of fused fabric layers. The presence of a TP polymer matrix in the composites lowered the Hyg value by 50%–67% compared to the same value for MM. However, in all the samples compared, the use of MIM had practically no effect on the basic Hyg values corresponding to comfortable wear conditions.

It is especially valuable that MIM makes it possible to regulate the rigidity of various parts of a composite garment-shaping piece without using multilayered laminated fabrics. This result can be achieved by using screens with changing printing patterns corresponding to the required reinforcement area ($S_{\text{A}}$) for screen printing of the modifying dispersion (see Table 3). In order to be able to regulate the rigidity, we derived mathematical dependencies describing changes in the main processing parameters of the composite depending on the $S_{\text{A}}$ value (with the first member of the equation right side reflecting the indicator value in the fused composites with a nonmodified interfacing fabric):

$$\begin{aligned} \text{EI}_{\text{K}}^{\text{MIM}} &= \text{EI}_{\text{K}}^{\text{FIM}} - 7.73 \times S_{\text{A}} + 76.84 \times S_{\text{A}}^2 \ (\text{range } \Delta \text{EI}_{\text{K}} = (6.7\text{--}27.4) \times 10^{-3} \ \text{N·cm}^2); \\ U_{\text{K}}^{\text{MIM}} &= U_{\text{K}}^{\text{FIM}} + 39.85 \times S_{\text{A}} + 19.72 \times S_{\text{A}}^2 \ (\text{range } \Delta U_{\text{K}} = 16.4\%\text{--}34.2\%). \end{aligned} \tag{6}$$

An analysis of the equations shows that by regulating $S_{\text{A}}$, it is possible to change the composite rigidity ($\Delta \text{EI}_{\text{K}}$) and elasticity ($\Delta U_{\text{K}}$) within a wide range of values.

### 3.3. Evaluation of Effects of Nanodispersed Fillers

One of the reinforcing nanomodifier types widely used in composite material production is detonation nanodiamonds (DND). Figure 7 shows the results of a comparison of the stress-strain properties of an MM1-based composite with those of standard FIM1 and its modified form with different amounts of DND introduced into the Akremos 120D oligoacrylate dispersion. The mechanically activated binary composition was deposited onto the interfacing material using screen No. 1 ($S_{\text{A}} = 0.35$).

Within the studied range of DND concentrations, the stiffness gain in the modified composite relative to the initial fused composite ($\Delta \text{EI}_{\text{DND}}$) was 1.7–2.6 times bigger than in the case of using oligoacrylate without a filler ($\Delta \text{EI}_0$), and the stiffness gain $\Delta U_{\text{K}}$ was 1.24–2.2 times bigger. The changes were more noticeable at the $C_{\text{DND}}$ up to 5 wt.%, when the $\Delta \text{EI}_{\text{K}}$ value doubled and $\Delta U_{\text{K}}$ increased 1.75-fold. A further increase in the DND concentration slowed down the indicator growth. This was probably caused by the negative DND particle surface zeta-potential. The higher modifier concentration increased the total charge value of the polymer dispersion and strengthened the forces of electrostatic repulsion with the negatively charged surface of the viscose fiber in the interlining.

The data in Figure 8 illustrate the limitations in the use of DND for modifying the interfacing polymer coating. The DND presence triggered agglomeration of the polymer dispersion particles, which, after 24-h storage, produced fractions 100–400 nm in size and micrometer fractions that took up more than 30% of the dispersed phase volume.

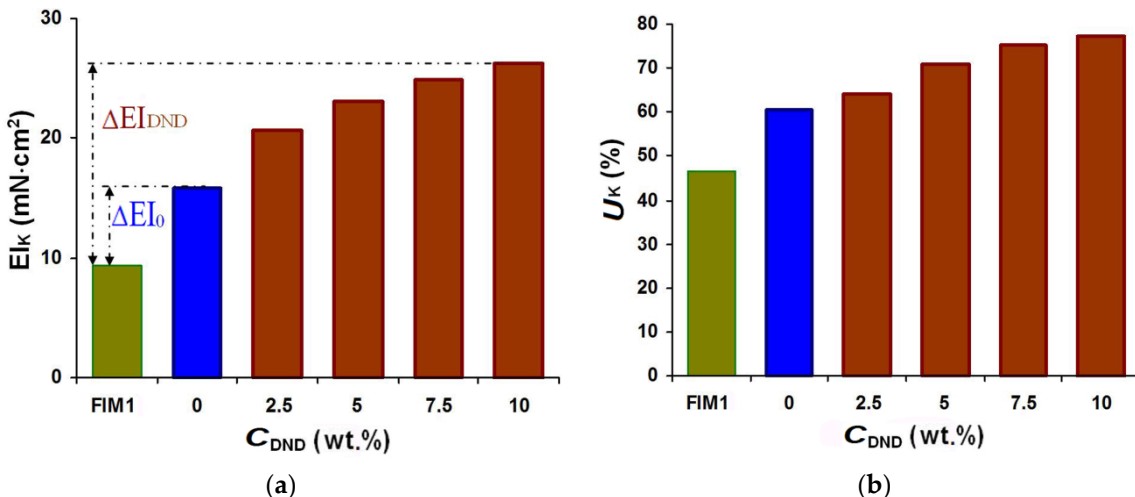

**Figure 7.** Dependencies of the composite stiffness (**a**) and elasticity (**b**) on the DND addition to the oligoacrylate dispersion ($C_{DND}$, wt.%).

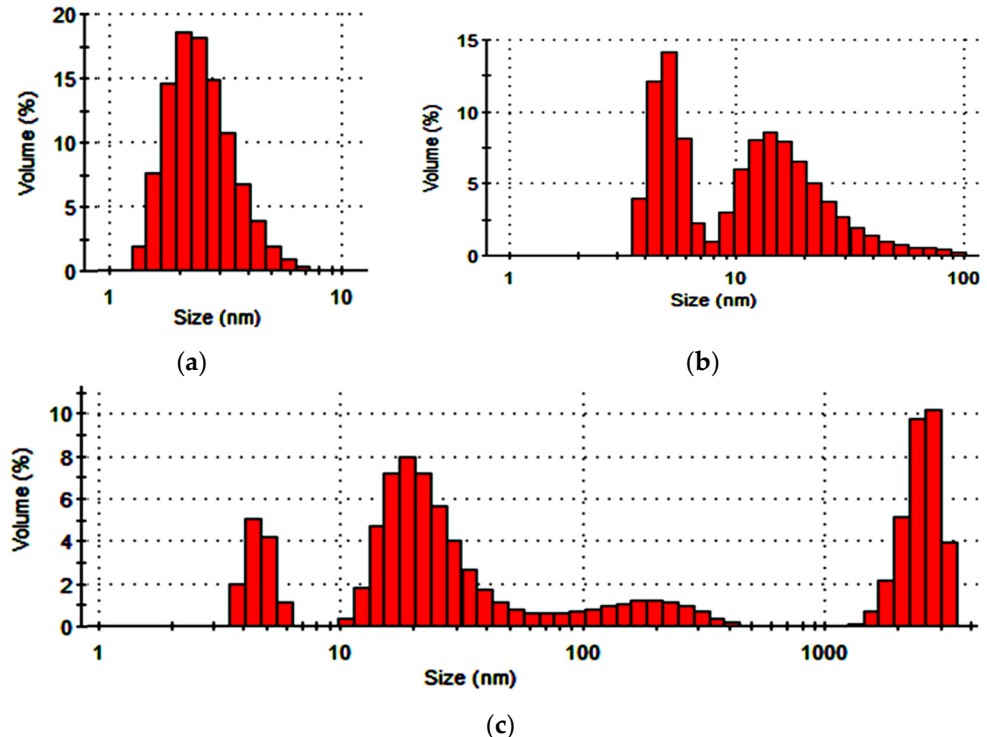

**Figure 8.** Particle size distribution of the dispersed phase relative volume in the DND preparation hydrosol (**a**) and its mixture with the Acremos 120D preparation after 1 h (**b**) and 24 h (**c**).

Since the polymer dispersion hydrosols are stable in colloidal state and the market prices of nanomodifiers are quite high, it is thought to be more effective to use available nanodispersed silicon dioxide (SD) compounds.

The data in Figure 9 illustrate the necessity of preliminary dispersion of colloidal SD compounds as the initial hydrosol form contained large aggregated fractions of 60–180 nm in size.

Ultrasound effects ensure hydrosol dispersion, with most of the volume occupied by fractions of individual SD grains capable of penetrating into the mesoporous spaces of the interlining fabric fibrous base. The SD ultradispersion effect was achieved by applying a combination of ultrasound, high-rate shear loads, and cavitation in the RPA apparatus. About 40% of the relative dispersed phase volume was taken up by the particles of less

than 10 nm in size. Evidently, mechanoacoustic effects led to crushing of the SD grains, which is illustrated by the micrographs with damaged spherical parts in Figure 10.

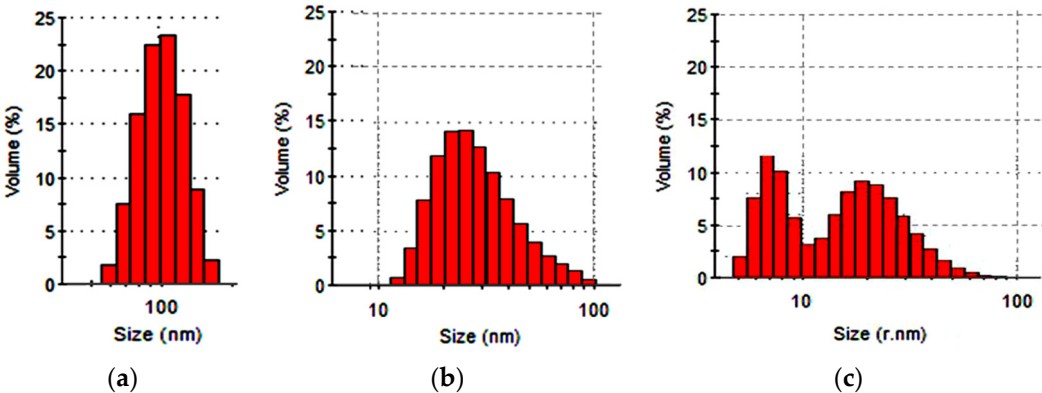

(a)  (b)  (c)

**Figure 9.** Changes in the fractional distribution of the initial SD hydrosol particles (**a**) after ultrasonic treatment (**b**) and mechanical activation in RPA (**c**).

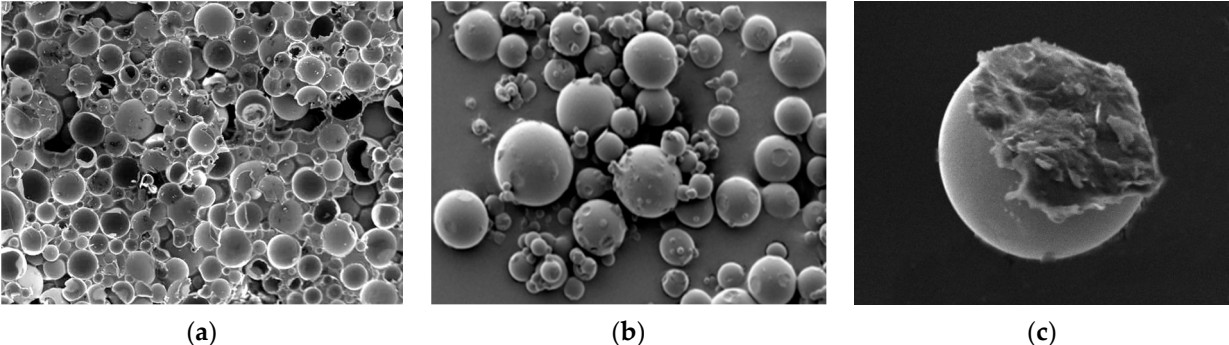

(a)  (b)  (c)

**Figure 10.** SEM images (**a**–**c**) of colloidal SD after mechanoacoustic processing in RPA.

The hypothesis about SD grain crushing caused by mechanical activation of the hydrosols and SD's ability to interact with oligoacrylates of different structures is confirmed by IR-spectroscopy studies [21]. The specifics of the interactions in the system are illustrated by the data in Figure 11.

The absorption bands for the SD compound reflected in curve 1 are reproduced in curve 2, which indicates that the chemical state of the colloidal $SiO_2$ remained unchanged after the US-treatment. Curve 2 also contained a group of oligoacrylate molecule bands: the stretching vibrations of the C=O groups in the butylacrylate fragments at 860 and 1692 $cm^{-1}$, the C–C stretching and skeletal vibrations at 1512 and 1600 $cm^{-1}$, as well as the bending (910 $cm^{-1}$) and stretching (2850 $cm^{-1}$) vibrations of the C–H bond in the alkyl chain. The high-intensity band appearing at 2156 $cm^{-1}$ on curve 2 was formed by the vibrations of the silanol groups participating in the formation of a hydrogen bond with the acrylate carbonyl. This indicates the physical nature of the adsorption interactions between the system components subjected to US-dispersion.

When the components were treated together in the RPA, it caused a considerable transformation of the spectrum (curve 3). Changes in the SD state were reflected in the lower intensity of the Si–O stretching vibration bands at 465, 630, 800, 1194 and 1960 $cm^{-1}$ and bending vibrations (at 486 and 560 $cm^{-1}$) in the O–Si–O bridge bonds, which indicated mechanically activated breakage of the siloxane bonds in the $SiO_4$ tetrahedron network. The higher intensity of the Si-OH bending vibrations peak at 870 $cm^{-1}$ indicated an increase in the number of silanol groups.

Polymerization triggered by mechanical activation of the binary system led to the disappearance of the vinyl group rocking vibration band on curve 3 (730 $cm^{-1}$) and peaks

of the $CH_2=C-$ double bond stretching vibrations (3044, 1721, 1550, 942 cm$^{-1}$) on curve 2. The band at 3044 cm$^{-1}$ on curve 3 shifted to the higher frequency region (3082 cm$^{-1}$) as a result of the increasing absorption of the stretching vibrations in the $CH_2$ and $CH_3$ groups formed by the oligoacrylate vinylidene unit transformation.

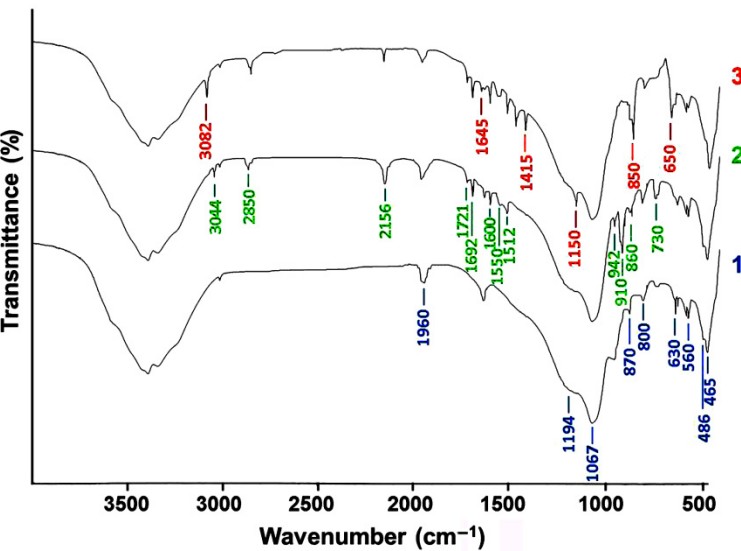

**Figure 11.** IR spectra of SD (1) and SD-Akratam AS compositions in a ratio of 7:3, obtained using ultrasound processing (2) and mechanical activation (3) of a binary hydrosol.

However, curve 3 had bands indicating the formation of new bond types. The band at 1150 cm$^{-1}$ demonstrated the appearance of a siloxane group bond with a Si–O–C carbon atom. The peaks at 650 and 850 cm$^{-1}$ indicated the formation of a Si–C bond. There were also absorption bands at 1415 and 1645 cm$^{-1}$ attributed to the scissoring vibrations of the hydrogen atoms in the Si–(R)C<(H)$_2$ groups. The results of the IR-spectra analysis allow us to conclude that subjecting the preparations simultaneously to the action of shear stress, ultrasound, and cavitation led to a break-up of the siloxane bonds in the structure of the silica dioxide nanoparticles and was accompanied by broken bond hydrolysis. At the same time, it is quite likely that Si–O–Si bond breakage may result in the formation of radical products, and the siloxane macroradical may get attached to the oligoacrylate "tail", which can facilitate stabilization of the binary hydrosol state and more uniform SD distribution within the polymer adhesive bulk.

Figure 12a presents the results of the thermal analysis of the two-component systems modelling copolymerization of oligoacrylate with a standard FIM polyamide adhesive.

The peak at 76.2 °C on curve 1 characterized the melting of the polyamide not interacting with the acrylate. Film formation started at 118.3 °C, whereas copolymerization in the presence of an initiator-halogenide complexonate of transition metals-and hardening of the adhesive completed at 186.7 °C. In the acrylate-SD composition (curve 2), thermally fusible polymer melting was contained, which made it possible to raise the drying temperature of the FIM being modified (see Figure 2) to $T_D$ = 85–90 °C. At the fusion stage, the heating must be carried out at temperatures not exceeding $T_G$ = 105–115 °C. Polyamide film and graft-copolymer formation processes represented one stage with a peak at 129 °C, which determines the minimum value of the $T_{WHT}$ parameter for the final WHT in order to shape the garment and stabilize its shape.

Figure 12b demonstrates the increase in the rigidity of the obtained composite as the $SiO_2$ content in the acrylate dispersion became higher. By changing the nanodispersed filler content from 5 to 25%, it was possible to increase the rigidity in composite garment shaping pieces 1.08–1.8 times, which may be a new solution to the problem of quick adjustment of the properties of interlining materials to the requirements for garment designs being developed.

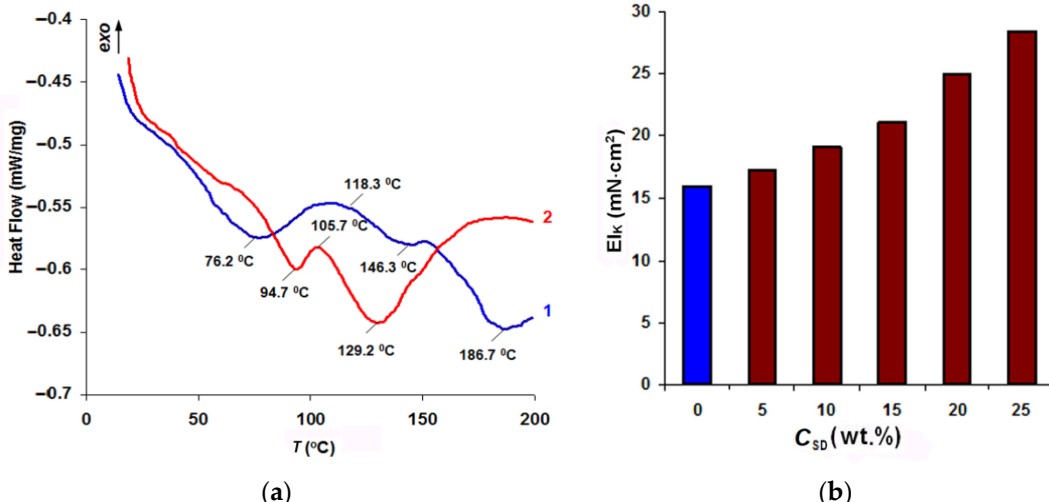

**Figure 12.** DSC analysis curves of PA-12AKR co-crystallization adducts of glue with the initial Akratam AS (1) and the acrylate-SD composition in a ratio of 3:1, (2) (**a**) and the stiffness index of the finished composite dependence on the SD concentration (**b**).

## 4. Discussion

The presented results indicate that a modified polymer coating ensured the formation of a graft-copolymer interface in the composite structure, which can have a great effect on the stress-strain properties of the garment piece made. The main difference of the developed MIM from traditional FIM was the replacement of the method consisting in forming 2D-structured adhesive layers between the fabrics to be joined to creating composites with a highly branched 3D interface. As Table 6 and Equations (2) and (5) show, the main technology used to make the rigidity of one type of the FIM textile base higher was increasing the adhesive dot area (TP mass), which was accompanied by a considerable reduction in the breathability of the finished composite. When MIM with a modified polymer coating were used, both the rigidity and the elasticity grew, which quite naturally led to higher strength and durability (see Equation (4)).

To demonstrate the effectiveness of using a modified coating, we calculated the main parameters of the fused fabric and finished composite obtained from MM1 fabric with FIM1 or MIM1 in garment pieces with the same preset value of the $EI_K$ indicator along the MM warp yarn. In case of FIM, we used Equation (2) to determine the necessary adhesive coating area $S_{TP}$. For MIM, we used Equation (5) to select the MP deposition area $S_A$. The results of the comparison of the calculated values for the fused fabric and finished composite are given in Table 8.

**Table 8.** Comparison of the fused fabrics and composites based on FIM and MIM characteristics calculated values.

| Required Value $EI_K$ ($10^{-3}$ N·cm$^2$) | Type of Interlining Material | Conditions for Creating $EI_K$ | | $A_{FF}$ (%) | $U_K$ (%) | $Q$ (dm$^3$/s m$^2$) |
|---|---|---|---|---|---|---|
| | | $S_{TP}$ (%) | $S_A$ | | | |
| 12 | FIM1 | 13.2 | - | 30.7 | 58.6 | 84.3 |
| 17 | FIM1 | 35.8 | - | 19.3 | 65.4 | 28.6 |
| | MIM1 | 13.2 | 0.31 | 30.7 | 72.8 | 84.3 |
| 22 | FIM1 | 60.3 | - | 15.3 | 72.8 | 11.5 |
| | MIM1 | 13.2 | 0.41 | 30.7 | 78.3 | 84.3 |
| 27 | FIM1 | 84.8 | - | 12.6 | 80.1 | 6.0 |
| | MIM1 | 13.2 | 0.5 | 30.7 | 83.5 | 84.3 |
| 32 | MIM1 | 13.2 | 0.56 | 30.7 | 87.1 | 84.3 |
| 40 | MIM1($C_{SD}$ 7.1%) | 13.2 | 0.31 | 30.7 | 86.7 | 84.3 |

It was found that the range of possible rigidity changes in the fabric with FIM1 as a result of increasing the TP amount was extremely limited. The increase of $5 \times 10^{-3}$ N·cm$^2$ in EI$_K$ could only be achieved if the coating area, $S_{TP}$, made up 35.8%; and the value growth of $10 \times 10^{-3}$ N·cm$^2$ could be only reached if the area occupied by the coating was 60.3%, which critically reduces the shaping ability and breathability. It should be said that the $A_{FF}$ reduction by less than 18% and the $U_K$ increase by more than 50% worsened the shaping conditions at the WHT stage and could cause a greater number of process defects in the form of creases and wrinkles on the shell fabric surface and quick shape relaxation. This means that it is possible to prepare composites with rigidity of $(20–40) \times 10^{-3}$ N·cm$^2$ by the traditional method only if several FIM layers are used.

The use of MIM made it possible to achieve the required value of the EI$_K$ indicator by modifying the interlining fabric at a minimum value of $S_{TP}$, even without reaching extremely high $S_A$ values. MIM-based composites have high values of all the parameters. This allows us to claim that MP deposition facilitates preparation of composites exceeding the traditional types in all the processing characteristics and consumer properties.

As an example, it was also shown that the maximum rigidity value ($39.8 \times 10^{-3}$ N·cm$^2$) that is required to ensure shape stability of the shoulder part of a large-size men's jacket of rigidly fixed form can be achieved by using an MP with a nanofiller. When the concentration of the additive, $C_{SD}$, in the composition with Akratam AS reached 7.1%, the target parameters of the composite could be obtained at the minimum values required for polymer coating modification ($S_{TP}$ and $S_A$).

Thus, the multifunctionality of the interfacing polymer coating was the result of the improvement of the processing characteristics and consumer properties of fused fabric and finished composite. The methods of screen or ink-jet printing with an aqueous MP dispersion made it possible to vary the composite rigidity by changing the $S_A$ value. The increase in the reinforcement area was accompanied by higher elasticity but did not lead to lower bond strength and did not reduce the shaping ability and breathability either. Changing the pattern of MP deposition over the MIM area made it possible to introduce gradient changes in the rigidity value within one reinforced garment piece. Higher reinforcement could be achieved by applying methods of nanostructural organization of MIM polymer coating. Easy-to-use methods of hydrosol ultradispersion and nanolayer surface modification of fibers were applied to ensure effective MP penetration into the intrafiber pore structure of textile bases.

MIM functionality could be widened by using nanodispersed fillers for fabric reinforcement or by providing special-purpose clothes with health improvement properties. An evaluation of the effect of the interface modification by ultradispersed silicon dioxide forms and detonation nanodiamonds showed that it was possible to increase the composite rigidity and elasticity. The higher elasticity (1.05–1.2-fold increase) in this case was more important, as it directly correlated with the resistance of a finished garment shape to wear.

It was found that an oligoacrylate graft dispersion could be used to stabilize the disperse state of nanosized fillers, to make their distribution in the composite structure more uniform and to achieve reproducible technological effects by modifying the interlining polymer coating. The proposed method of simultaneous mechanical activation of a nanodispersed filler with an MP hydrosol could be applied to immobilize highly coercive nanoparticles, strontium ferrite in particular, in a fibrous material. The effectiveness of the developed method in producing magnetic fabrics generating a constant low-intensity magnetic field in the near-surface layer is shown in [13]. Special garment design [14] improves the adaptation and regeneration capacity of healthy people in psychologically or physically stressful situations and makes it possible to apply magnetic therapy for treatment and prevention of many diseases.

## 5. Conclusions

The presented results indicate that the technology proposed by us for modifying polymer coatings of fusible interlining materials by depositing a grafted oligomer dispersion

capable of penetrating into the intrafiber pore structure of the textile base can be used to widen the functionality of interlining materials and composite garment pieces based on them. The nanostructured architecture of the modified polymer coating and timely (at the final WHT stage) oligoacrylate copolymerization with the macromolecules of a thermoplastic adhesive facilitate the formation of highly branched 3D-structured interfaces in the composites, which considerably increases the processing characteristics and consumer properties of the finished garments in comparison with those made with standard FIM.

The studied methods of ultradispersing polymer dispersions and nanolayer fibrous substrate surface modification complement each other and make it possible to control the area of formation of a highly developed adhesive structure and to reduce the amount of reinforcing nanomodifiers to reasonable values. Together with screen printing methods, in which it is possible to change the area of modifying composition deposition, they allow for the development of a wide variety of interlining fabrics from a small range of standard FIM and quick adjustment of their functional characteristics in order to produce clothes of any shape and with any elasticity degree.

## 6. Patents

Patent RU 2,383,672 C2. Compound for giving stability of shape to apparel components. Publ. 10.03.2010.

**Author Contributions:** Conceptualization, N.K. and S.K.; methodology, S.K.; validation, A.B., E.N. and N.K.; formal analysis, O.R. and S.A.; writing—original draft preparation, N.K. and S.K.; writing—review and editing, A.B.; visualization, O.R. and S.A.; supervision, A.B.; project administration, E.N. All authors have read and agreed to the published version of the manuscript.

**Funding:** This research received no external funding.

**Institutional Review Board Statement:** Not applicable.

**Informed Consent Statement:** Not applicable.

**Data Availability Statement:** Data is contained within the article.

**Conflicts of Interest:** The authors declare no conflict of interest.

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
