# Peer review of "Multifunctional Polymer Coatings of Fusible Interlinings for Sewing Products"

_coatings, doi:10.3390/coatings11060616_

Round 1

Reviewer 1 Report

Multi-functional polymer coatings of fusible interlinings for sewing products by Kornilova et al. describes a method for the modification of fusible interlining materials using functional polymer dispersions and their interactions with thermoplastics.  The experimental work is interesting, and the presentation is of good quality and therefore I believe that this manuscript should be published in Coating after few modifications:
The experimental part must be described in further detail, including the experimental conditions as well as the analysis methods. This is the case of all experimental methods used here and particularly for DSC and DLS. 
For example, the size distribution from DLS and particularly from zetasizer nano is highly sensitive to the analysis method, especially for such polydisperse particles. For example, the mean values of figures 3,4, 8, and 9 could be correct but the width and the shape of the distributions are model dependent. The authors should clarify this in their manuscript or present only the mean values from the first cumulant.

Author Response

Response to Reviewer 1 Comments

Point 1: The experimental part must be described in further detail, including the experimental conditions as well as the analysis methods.

Response 1: The experimental part in the question of determining the conditions for the interaction of polymers and the modes of obtaining modified interlining materials for clothes is described in sufficient detail in the previous articles of the authors (indicated in the list of cited literature):

17 - Kornilova, N.L., Koksharov, S.A, Arbuzova, A.A. et al. Development of reinforced interlining materials to regulate elastic properties. Indian J. of Fibre & Textile Res. 2017, 42, 150-159. http://nopr.niscair.res.in/handle/123456789/42069

16 - Koksharov, S.A., Kornilova, N.L., Fedosov, S.V. Development of reinforced composite materials with a nanoporous textile substrate and a brush-structured polymer interfacial layer. Russ. J. Gen. Chem. 2017, 87, No.6, 1428-1438. https://doi.org/10.1134/S1070363217060469

15 - Koksharov, S.A., Kornilova, N.L., Shammut, Ju.A. et al. Synthesis of a highly chained polymeric connecting in the structure of a multilayered package for garments. Key Engineering Materials 2019, 816, 219-227. https://doi.org/10.4028/www.scientific.net/KEM.816.219

5 - Koksharov, S.A., Kornilova, N.L., Shammut, Yu.A. Design of composite materials for clothes. MATEC Web of Conferences 2020, 315, 03001. https://doi.org/10.1051/matecconf/202031503001

The experimental part of the methods used for assessing the properties of the obtained interlining materials is described in sufficient detail in the article [17].

Point 2:

This is the case of all experimental methods used here and particularly for DSC and DLS. 
For example, the size distribution from DLS and particularly from zetasizer nano is highly sensitive to the analysis method, especially for such polydisperse particles. For example, the mean values of figures 3,4, 8, and 9 could be correct but the width and the shape of the distributions are model dependent. The authors should clarify this in their manuscript or present only the mean values from the first cumulant.

Response 2.

The size of nanoparticles in hydrosols was measured by dynamic light scattering on a Zetasizer Nano ZS analyzer; the signal accumulation time in a series of three measurements was 20 min. The analysis of the measurement results was carried out by an automated program based on the solution of the Fredholm integral equation of the first kind with an exponential kernel for the normalized correlation function [add. link]. To increase the recording ability of the measuring system, taking into account the recommendations [19] for the study of polyfraction systems in the results processing window, the value of the Lower Threshold indicator "0.05" must be corrected to "0".

[add. link] - Yan Y.D., Clarke J. H. R. In-situ determination of particle size distributions in colloids. Advances in Colloid and Interface Science. 1989. V. 29. P.277-318. https://doi.org/10.1016/0001-8686(89)80011-9

Reviewer 2 Report

For the future work I recommend characterization of materials using internationally popular methods, as presented in the reference 2 (Polymers 2018, 10, 1230) "Kawabata Evaluation System of Fabric (KESF) is a popular fabric objective measurement  system.   Here  low-stress  mechanical properties  include  tensile,  bending,  shearing,surface and compression. Total 16 low-stress mechanical properties can be tested.

Line 88

comb-like structures with  numerous side radicals attached to the backbone CORRECT comb-like structures with numerous side chains attached to the backbone

Line 133

ultrasound treatment in a UZDN-2T disperser at a frequency of 22 Hz REMARK : frequency value too low, maybe 22 kHz?

Line 239

polyvinyl chlorides, polyurethanes, polyvinyl acetates CORRECT polyvinyl chloride, polyurethanes, polyvinyl acetate

Author Response

Response to Reviewer 2 Comments

Point 1: For the future work I recommend characterization of materials using internationally popular methods, as presented in the reference 2 (Polymers 2018, 10, 1230) "Kawabata Evaluation System of Fabric (KESF) is a popular fabric objective measurement  system.   Here  low-stress  mechanical properties  include  tensile,  bending,  shearing, surface and compression. Total 16 low-stress mechanical properties can be tested.

Response 1: Thank you for this suggestion. We would like to use this Evaluation System of Fabric, but unfortunately we do not have access to equipment. We would be glad to do joint research with an organization that has different machines on which tests can be performed.

Point 2:

Line 88

‘comb-like structures with  numerous side radicals attached to the backbone’ CORRECT -‘comb-like structures with numerous side chains attached to the backbone’

Line 239

‘polyvinyl chlorides, polyurethanes, polyvinyl acetates’ CORRECT- ‘polyvinyl chloride, polyurethanes, polyvinyl acetate’

Response 2. Thank you, the changes have been made in the text.

Point 3:

Line 133

ultrasound treatment in a UZDN-2T disperser at a frequency of 22 Hz REMARK : frequency value too low, maybe 22 kHz?

Response 3. Thank you, the changes have been made in the text - 22 kHz.

Reviewer 3 Report

The paper deals with the application of a modified fusible interlining material to textiles. The appication of a tridimensional, branched polymerical structure should improve the desired properties, enhancing the adhesion without a lost of breathability or comfort. Results are interesting and presented in an effective way. 

Just a question:  

Did authors tested the fastness of the treatment to washing or rubbing? Could they find any improvement due to the branched 3D structure fo the polymer, with respect to the 2D structure?

The presented topic is of interest in the field of textile treatments, presenting a quite new approach to FIM application on fibers. The paper is clear and well written, in good English. Maybe you should just check for some refuse in the text. Materials and methods are fully described. Characterizations are complete and well commented, supported by clear figures and tables.
Conclusions are consistent with obtained results and with the aim of the study. The cited bibliography is quite recent and appropriate.

In conclusion I recommend to publish the article as it is.

Author Response

Response to Reviewer 3 Comments

Point 1:

Did authors tested the fastness of the treatment to washing or rubbing?

Response 1: Thank you for the question. We did not test the fastness of the treatment to washing or rubbing, because we tested the combination of materials used in the men's jacket. Traditionally, a men's jacket is dry cleaned, and the dry cleaning resistance measurements are shown in Table 7 (SSdry clean).

Point 2: Could they find any improvement due to the branched 3D structure for the polymer, with respect to the 2D structure?

Response 2: It is likely that the fastness of the treatment to washing or rubbing can be improved due to the branched 3D structure of the polymer compared to the 2D structure. We will definitely conduct research on these properties in the future.

Point 3:  Maybe you should just check for some refuse in the text.

 Response 3:  Thanks for the advice. We have checked the text again and believe that there is no unnecessary material in it. We believe that everything is important.

Round 2

Reviewer 1 Report

The authors have made some of the required modifications but did not go into the details of the DLS analysis. However, with the given experimental descriptions the experiment can be reproduced. Therefore I recommend the publication of this manuscript in Coating.